# SDGH-Net: Ship Detection in Optical Remote Sensing Images Based on Gaussian Heatmap Regression

**Zhenqing Wang [1,2], Yi Zhou [1,*], Futao Wang [1], Shixin Wang [1] and Zhiyu Xu [1,2]**

1  Aerospace Information Research Institute, Chinese Academy of Sciences, Beijing 100094, China; wangzhenqing19@mails.ucas.ac.cn (Z.W.); wangft@aircas.ac.cn (F.W.); wangsx@aircas.ac.cn (S.W.); xuzhiyu@aircas.ac.cn (Z.X.)
2  University of Chinese Academy of Science, Beijing 100049, China
*  Correspondence: zhouyi@aircas.ac.cn; Tel.: +86-010-64879460

**Abstract:** The ship detection task using optical remote sensing images is important for in maritime safety, port management and ship rescue. With the wide application of deep learning to remote sensing, a series of target detection algorithms, such as faster regions with convolution neural network feature (R-CNN) and You Only Look Once (YOLO), have been developed to detect ships in remote sensing images. These detection algorithms use fully connected layer direct regression to obtain coordinate points. Although training and forward speed are fast, they lack spatial generalization ability. To avoid the over-fitting problem that may arise from the fully connected layer, we propose a fully convolutional neural network, SDGH-Net, based on Gaussian heatmap regression. SDGH-Net uses an encoder–decoder structure to obtain the ship area feature map by direct regression. After simple post-processing, the ship polygon annotation can be obtained without non-maximum suppression (NMS) processing. To speed up model training, we added a batch normalization (BN) processing layer. To increase the receptive field while controlling the number of learning parameters, we introduced dilated convolution and added it at different rates to fuse the features of different scales. We tested the performance of our proposed method using a public ship dataset HRSC2016. The experimental results show that this method improves the recall rate of ships, and the F-measure is 85.05%, which surpasses all other methods we used for comparison.

**Keywords:** ship detection; optical remote sensing images; Gaussian heatmap; SDGH-Net; deep learning

## 1. Introduction

As a large-scale, long-distance ground detection technology, remote sensing can quickly collect information on the ground and at sea by acquiring remote sensing images of regions of interest. Ships play an important role as a means of transportation or operation at sea. Ship detection in optical remote sensing images is a challenging task and has a wide range of applications in ship positioning, maritime traffic control, and ship rescue [1].

Early ship detection is generally achieved using synthetic aperture radar (SAR) images [2–4]. SAR images are not affected by weather such as clouds and fog, and have strong anti-interference ability and strong penetration. However, SAR images still have disadvantages, such as the limited number of SAR sensors, a relatively long revisit period, and a relatively low resolution. With the increase in the number of optical sensors and the resulting improvement in the continuous coverage of optical sensors, more studies have examined ship detection based on optical remote sensing images.

Traditional optical remote sensing image ship detection generally includes two steps: candidate area extraction and classification [5,6]. Firstly, the candidate area of the ship object is obtained according to the inherent characteristics such as the scale and shape of the ship object or the visual attention mechanism, and then the corresponding features of the candidate area are extracted for training to obtain the detection result. However, due to

the lack of high-dimensional semantic information in traditional methods, the accuracy of the detection model is low.

In recent years, convolutional neural networks (CNNs) in the field of deep learning have performed well in the fields of image classification, semantic segmentation, and target detection [7–9]. From the perspective of ideas, deep learning algorithms for target detection can be divided into two categories. One is the two-stage method; that is, the process is divided into two parts: generating candidate frames and identifying objects in the frame. The second is the one-stage method, which unifies the entire process and directly provides the detection results. The two-stage method was first developed in 2014. The regions with CNN features (R-CNN) method introduces deep learning convolution neural networks into the field of target detection, which considerably improves the accuracy of target detection, but it also has the disadvantages of the region proposal selection algorithm being time-consuming, repeated calculation of overlapping area features, and relatively large time and memory consumption [10]. With the introduction of spatial pyramid pooling networks (SPP-net) [11], fast R-CNN [12], faster R-CNN [13], and other algorithms, the structure of the two-stage model has become more complete. For the pursuit of speed, the one-stage method began to be developed a year after R-CNN was proposed. You Only Look Once (YOLO) did not show the process of obtaining the region proposal, and unified the extraction of the region proposal into a regression problem, which improved the detection speed [14]. Subsequently, single shot multibox detector (SSD) [15], YOLOv2 [16] and YOLOv3 [17] were proposed successively. A large part of YOLO's improvement in each generation is determined by the improvement in the backbone network, from v22019s darknet-19 to v3's darknet-53.

R-CNN, YOLO and other series of networks have been proven to be effective in natural image target detection, so remote sensing researchers are applied these networks to optical remote sensing image ship detection. Feng et al. constructed a rotation-based network with a sequence local context (SLC) module and a horizontal region of interest (HRoI) pooling layer, which can accurately generate rotated bounding boxes, thereby achieving a higher recall rate [18]. Ma et al. predicted the angle range of ships by constructing a ship-oriented classification network. Within the predicted angle range, the angle difference between the area proposal and the ground truth box can be limited to a smaller value, thereby obtaining a more accurate area proposal [19]. Tian et al. described a nearshore ship detection framework based on multi-scale feature pyramid fusion network, rotating region suggestion network and interest pool context rotating region [20]. To overcome the limitations of dense distribution and different scales, Guo et al. proposed adding the balanced feature pyramid (BFP) module and the intersection over union (IoU)-balanced sampling (BS) module rotational Libra R-CNN (R-Libra R-CNN) [21]. Wu et al. proposed a coarse-to-fine ship detection network (CF-SDN), which included a sea-land separation algorithm, coarse-to-fine ship detection network and multi-scale detection strategy [22]. Zhang et al. proposed a ship detection model including feature fusion backbone, recall-priority branch, precision priority branch, and priority-based selection (PBS) module, which reduced the number of false alarms and missing ships [23]. Tang et al. proposed a single-shot ship detection approach based on region of interest preselected network. To efficiently match ship candidates, the principle of their approach is to distinguish suspected areas from the images based on hue, saturation, and value (HSV) differences between ships and the background [24].

The above-mentioned target detection algorithms use the fully connected layer to directly return the coordinate points. The advantage of this method is that the output is the coordinate point, the training and forward speed are relatively fast, and it is an end-to-end total differential training. However, it lacks spatial generalization ability; that is, it loses the spatial information on the feature map. For the fully connected layer, Lin et al. stated in 2014 that the fully connected layer easily leads to overfitting, thereby reducing the generalization ability of the entire network [25]. The weight of the fully connected layer depends heavily on the distribution of the training data, so the fully connected layer

relatively easily causes overfitting. The overall process of Gaussian heatmap regression is relatively simple, and its framework is a fully convolutional neural network consistent with image segmentation. However, unlike image segmentation, the ground truth of the Gaussian heatmap regression is a Gaussian heatmap ranging from zero to one. Gaussian heatmap regression has no fully connected layer, and the output feature map has a large size and strong spatial generalization ability. In addition, using heatmap to supervise is an easier way to learn than regressing coordinates. The network does not need to convert the spatial position into coordinates by itself. Therefore, more tasks [26–28] use fully convolutional networks based on Gaussian heatmap regression for target detection.

We designed SDGH-Net, which innovatively applies Gaussian heatmap regression method to ship detection in optical remote sensing images. SDGH-Net only outputs one channel, which is the feature map of the ship area. With the help of post-processing algorithms, it can effectively detect ships in optical remote sensing images. In SDGH-Net, the Gaussian heatmap generated by the ship enclosing polygon is used as the ground truth. SDGH-Net is a variant of the fully convolutional neural network U-Net [29]. After each convolution operation, we add batch normalization processing to improve the network training speed. To increase the receptive field while controlling the number of parameters, we introduce dilated convolution and fuse the dilated convolution with different rates to adapt to multi-scale features. To overcome the imbalance of positive and negative samples, we use the weighted mean square error as the loss function of the model. After obtaining the feature map of the ship area, the final ship enclosing polygon can be obtained through a simple post-processing algorithm. Because there is no redundant prediction box, we do not need to perform non-maximum suppression (NMS) processing.

The experimental results on the public ship dataset HRSC2016 [30] verify the effectiveness of this method. This method has strong robustness to ship targets of various sizes and types, and has excellent performance in precision, recall and F-measure standard. The main contributions of this paper are as follows:

- To solve the overfitting problem of the fully connected layer, a novel and concise SDGH-Net model of ship detection in optical remote sensing image based on Gaussian heatmap regression is proposed. The model directly outputs the feature map of the ship, and the position of the ship can be obtained after simple post-processing, which improves the ship detection capability.
- To more accurately detect targets of different scales, we designed a multi-scale feature extraction and fusion network to construct multi-scale features using dilated convolution, which helps to improve the capability to detect different ships.

The code in this work will be updated to GitHub (https://github.com/WangZhenqing-RS/SDGH-Net-Ship-Detection-in-Optical-Remote-Sensing-Images-Based-on-Gaussian-Heatmap-Regression) in the near future.

The rest of this article is organized as follows. The second section explains how the Gaussian heatmap ground truth is generated, and introduces our SDGH-Net model in detail. The third section provides different comparative experiments and their corresponding results. The fourth section analyzes the different results. Finally, the fifth section summarizes the full text.

## 2. Materials and Methods

Here, we propose the optical remote sensing image ship detection network SDGH-Net based on Gaussian heatmap regression. SDGH-Net is an improved U-Net model; the overall framework is shown in Figure 1. On the left is the contraction path used to extract high-dimensional feature information, and on the right is the symmetrical expansion path for precise positioning. Different from the conventional ship detection model, the ground truth of the model is a Gaussian heatmap with the same length and width as the input image. To improve the network training speed and accelerate the convergence process, the model has batch normalization (BN) added to each layer of the network after the

convolution operation. To further extract features of different scales, the model introduces a dilated convolution mechanism.

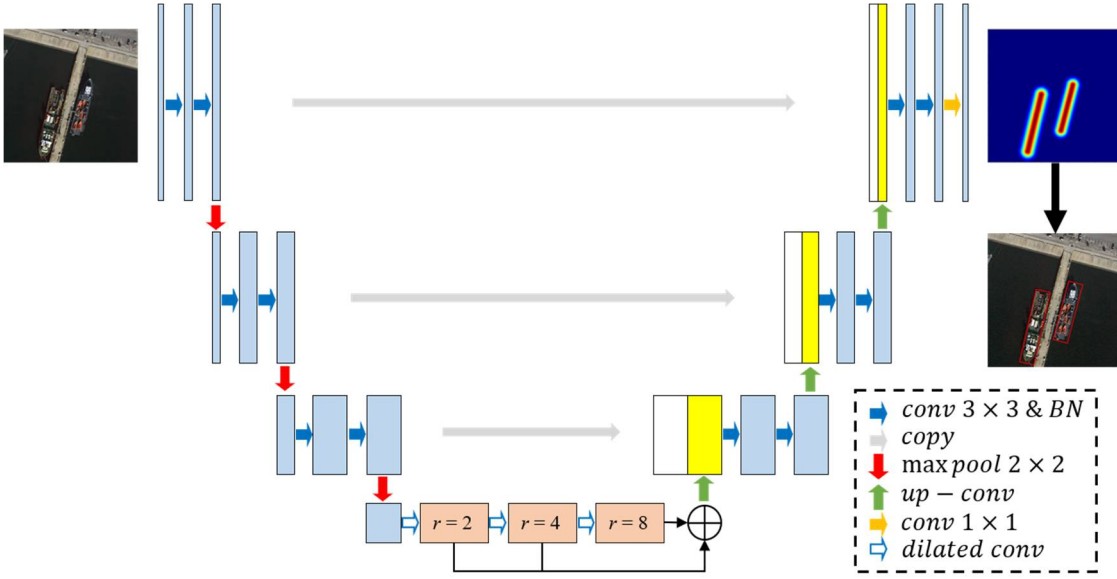

**Figure 1.** Overall framework of SDGH-Net.

### 2.1. Ground Truth Generation

Similar to the ground truth value of image segmentation, the ground truth value of this method also has the same size as the input remote sensing image. However, the value of the pixel of the ground truth value is a probability, which represents the probability that the pixel belongs to the ship area. The closer the pixel is to the center of the ship area, the closer the probability value is to 1.0, and vice versa for the closer it is to 0.0. Detailed information is shown in Figure 2.

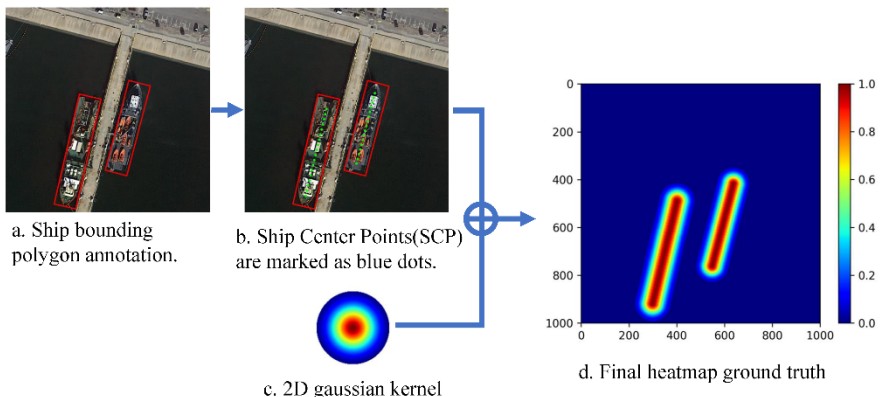

**Figure 2.** How the Gaussian heatmap ground truth is created.

First, according to the ship polygon annotation, the center line of the long side of the ship area can be obtained. On the center line, a series of ship center points (SCPs) {SCP1, SCP2, ..., SCPn} separated by a certain step length S can be obtained, as shown in Figure 3. Secondly, for each SCP, a 2D Gaussian kernel is used to generate a corresponding Gaussian heatmap with a radius of *R*, and then fused to obtain the final Gaussian heatmap ground truth. The radius *R* is defined as:

$$R = \frac{dis(V_0, V_1) + dis(V_2, V_3)}{4} \qquad (1)$$

where $(V_0, V_1)$ and $(V_2, V_3)$ are the respective endpoints of the two short sides of the ship polygon annotation, that is, $R$ is expressed as half of the average length of the two short sides.

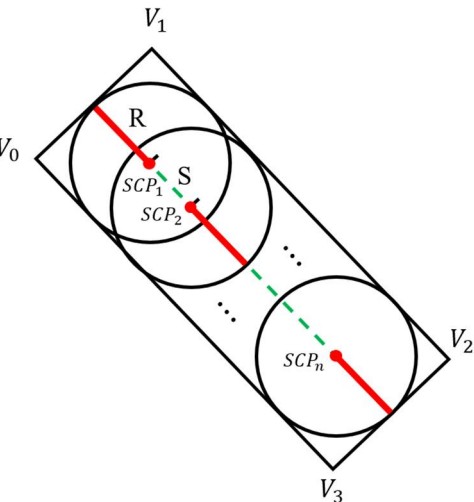

**Figure 3.** How ship center points (SCPs) are selected.

To avoid using part of the background as a true value and introducing noise, the first and last two SCPs need to have a distance of $R$ from the two points at the beginning and the end of the center line, respectively.

*2.2. Improved Network Structure*

2.2.1. Batch Normalization

For a deeper neural network like SDGH-Net, if the data distribution of the first few layers of the network changes slightly, the later layers will accumulate and enlarge. Once the input data distribution of a certain layer of the network changes significantly, the SDGH-Net model needs to learn the new data distribution and then update the parameters, which will greatly reduce the network training speed. To solve this problem, we added a batch normalization processing method after each convolution. BN is a strategy proposed by Ioffe and Szegedy where, when a certain layer of the network is input, the normalization process is performed before entering the next layer of the network [31]. The normalization layer here is a learnable and parameterized network layer, expressed as:

$$y^{(k)} = \gamma^{(k)}\overline{x}^{(k)} + \beta^{(k)} \tag{2}$$

where $y^{(k)}$ is the batch normalization result of the k-th layer, $\overline{x}^{(k)}$ is the standard deviation normalization result, and both $\gamma^{(k)}$ and $\beta^{(k)}$ are learning parameters. We added a BN layer after each $3 \times 3$ convolution operation.

2.2.2. Dilated Convolution

In the CNN model, more contextual information is usually obtained by expanding the receptive field, which is mainly achieved by increasing the filter size or using a pooling layer. However, increasing the filter size will greatly increase the number and complexity of the model's parameters, and too many pooling layers will result in the loss of spatial location information. Recently, dilated convolution [32] has become popular because compared with traditional convolution, it can effectively expand the receptive field while maintaining the relative spatial position of the filter size and feature map.

Figure 4a corresponds to a $3 \times 3$ dilated convolution with a rate of 1, which is the same as the ordinary convolution operation. Figure 4b corresponds to a $3 \times 3$ dilated convolution with a rate of 2. The actual convolution kernel size is still $3 \times 3$, but for a $7 \times 7$ image patch, only 9 red pixels and $3 \times 3$ convolution kernels are convoluted, and the remaining points

are ignored. Although the convolution kernel is only $3 \times 3$ in size, the receptive field of this convolution has increased to $7 \times 7$. The subgraph (c) is a dilated convolution with rate 4, which can reach a receptive field of $15 \times 15$. The receptive field of dilated convolution increases exponentially, while the number of parameters increases linearly.

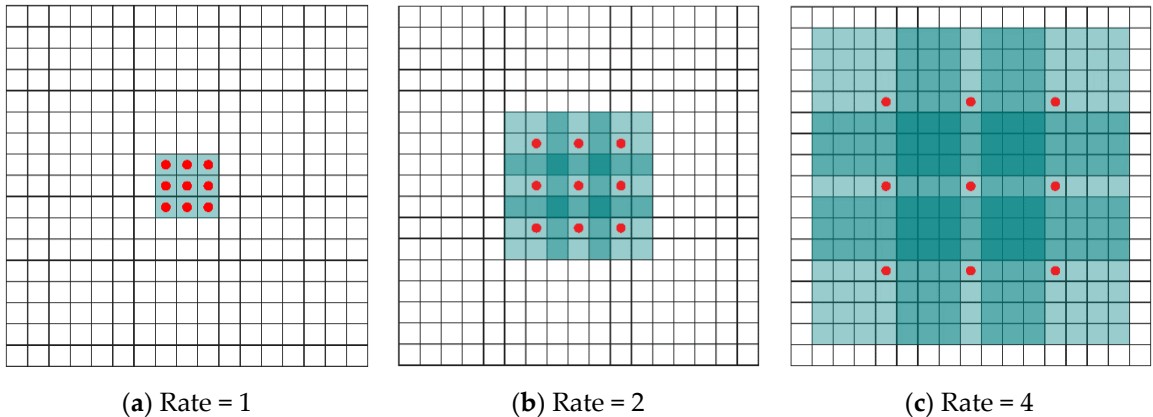

(**a**) Rate = 1　　　　　　　　　　(**b**) Rate = 2　　　　　　　　　　(**c**) Rate = 4

**Figure 4.** Receptive field size of different dilated convolutions with rates of 1, 2, and 4. (**a**) Rate = 1, (**b**) Rate = 2, (**c**) Rate = 4.

To increase the receptive field while limiting parameter growth, we introduced a dilated convolution mechanism in the fourth level of the model, and performed continuous dilated convolutions with rates of 2, 4, and 8 on the previously acquired features. In addition, to further extract features of different scales, we merged the results of the three dilated convolutions in an additive form.

2.2.3. Loss Function

Mean squared error (*MSE*) is commonly used as the loss function of regression algorithms. Since the ratio of ships (positive samples) and background (negative samples) in our remote sensing images is quite different, to compensate for this imbalance, we use weighted *MSE*:

$$MSE = \frac{1}{N} \sum_{i=1}^{n} \left( y_{pred} - y_{true} \right)^2 \tag{3}$$

$$MSE_{Weight} = \frac{N_{ship}}{N} MSE(BG) + \frac{N_{BG}}{N} MSE(ship) \tag{4}$$

where $y_{pred}$ represents the pixel value output by SDGH-Net, $y_{true}$ represents the pixel value of the ground Gaussian heatmap, $N$ is the total number of pixels in the Gaussian heatmap, *MSE(BG)* represents the mean square error of the background area, and *MSE(ship)* represents the ship. The mean square error of the area $N_{ship}$ represents the total number of pixels in the ship area of the Gaussian heatmap, and $N_{BG}$ represents the total number of pixels in the background area of the Gaussian heatmap.

*2.3. Post-Processing*

Because the SDGH-Net prediction result is a ship feature map, to obtain the ship polygon annotation, we need to use a post-processing method to extract the final polygon annotation from the ship feature map. If we directly take an area greater than 0 in the ship feature map as the ship area, it may cause adhesion between close-range ship groups, and cause multiple ships to be assigned only one polygon annotation. To overcome this problem, we applied a simple treatment. As shown in Figure 5, we performed threshold processing on the ship feature map; that is, the area larger than the threshold $T_{main}$ was set as the main area of the ship, and then the main area was expanded to obtain the entire area of the ship. Then, the boundary rectangle of the entire area of the ship was the boundary polygon we needed. In this experiment, the value of $T_{main}$ was set to 0.6.

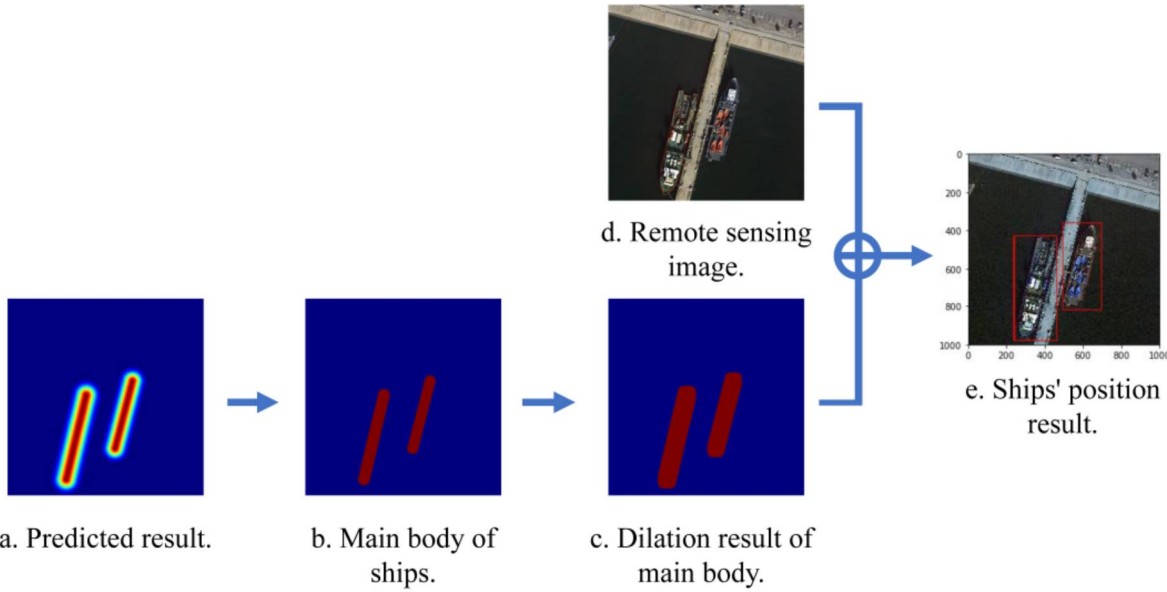

**Figure 5.** Post-processing.

## 2.4. Training

### 2.4.1. Data Augmentation

Data augmentation is an indispensable step in deep learning image processing tasks. It refers to the process of performing some transformation operations on training sample data to generate new data. The fundamental purpose of data augmentation is to obtain sufficient sample data, avoid over-fitting during model training, enhance the generalization ability of the model, and accelerate model convergence.

For remote sensing images, since the imaging process of the sensor capturing the same object at different angles will show different positions and shapes on the image, the transformed samples can make it easier for the model to learn the features of the object's rotation invariance to better adapt to different forms of images. Therefore, we performed geometric transformation (including horizontal flip, vertical flip and diagonal flip) data augmentation operations on the training data, as shown in Figure 6.

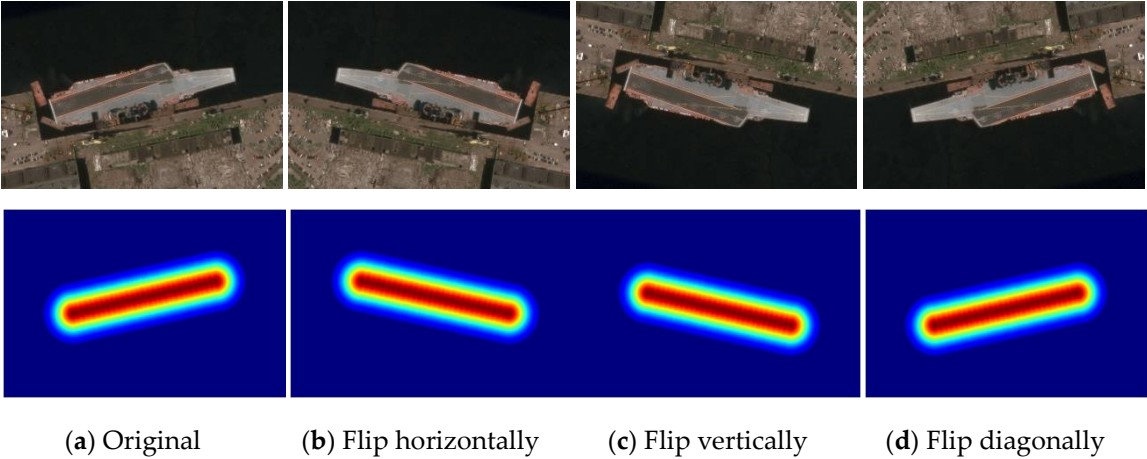

(**a**) Original      (**b**) Flip horizontally      (**c**) Flip vertically      (**d**) Flip diagonally

**Figure 6.** Data augmentation. The top row is the remote sensing image, and the bottom row is the ground truth. (**a**) Original, (**b**) Flip horizontally, (**c**) Flip vertically, (**d**) Flip diagonally.

### 2.4.2. Adjust Learning Rate Dynamically

As an important hyperparameter in deep learning, the learning rate determines whether the objective function can converge to a minimum and when it converges to the minimum. When the learning rate is too low, the convergence process will become very slow; otherwise, the gradient may oscillate back and forth near the minimum value, and may even cause convergence failure. A proper learning rate can make the objective function converge to a local minimum in a reasonable time. We used a dynamic adjustment strategy to optimize the learning rate. The initial learning rate was set to 0.0001, and when the loss of the validation set was no longer improved for five consecutive epochs, the learning rate was divided by 5. This can improve the efficiency of model training by jumping out of the local minimum to find the best solution in the shortest time.

### 2.4.3. Early Stopping

The number of model training times is also an important parameter in deep learning. If the number of training times is too low, the model cannot be fully learned; if the number of training times is too large, the model will learn too much, and this can result in overfitting [32]. Therefore, we adopt the strategy of early termination, and the maximum number of epochs was set to 100. When the validation set loses 30 consecutive epochs and no longer improves, the training was interrupted. In this way, we can avoid over fitting the model and obtain the best model in the training process.

### 2.5. Testing

Test Time Augmentation

Test time augmentation (TTA) is a common method used to improve accuracy. By enhancing the test data set and averaging the enhanced test results to obtain the final prediction result, the accuracy and reliability of the classification result can be increased. In this study, the test samples were enhanced by horizontal flipping, vertical flipping, and diagonal flipping, and then input into the trained model. The enhanced test data were calculated four times obtain get the average output as the final prediction result of the image, as shown in Figure 7.

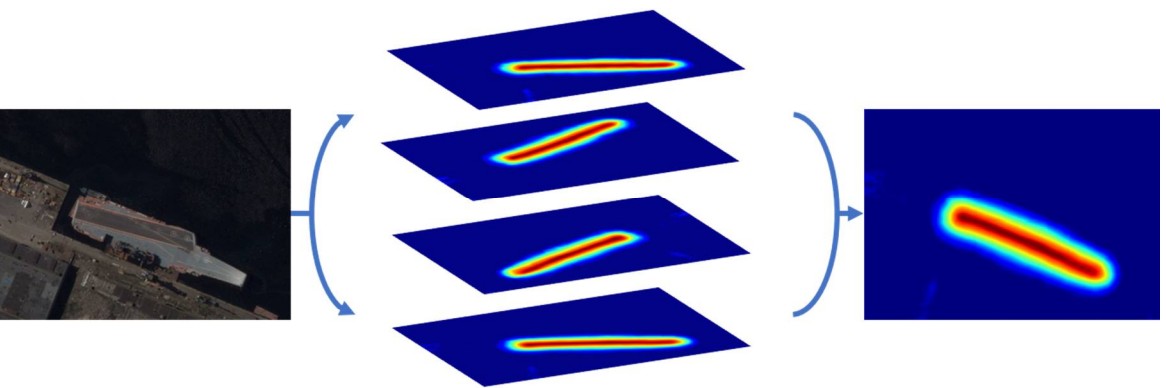

**Figure 7.** Test time augmentation.

## 3. Experiments and Results

To evaluate the performance of our proposed SDGH-Net model, we compared it with other methods. The experimental settings introduced in this section include data sets, evaluation indicators, and comparison methods.

### 3.1. Dataset

The ships in the HRSC2016 [30] dataset include ships on sea and ships close inshore. There are a total of 1061 images including 70 sea images with 90 samples and 991 sea-land

images with 2886 samples. All images were collected from five ports in Google Earth. The five ports are Everett, Newport-Rhode Island, Mayport Naval Base, Norfolk Naval Base, and Naval Base San Diego. The resolution of the images is between 0.4 and 2 m, and the size of the images is between $300 \times 300$ and $1500 \times 900$, most of which are larger than $1000 \times 600$. The data set was divided into the training set, validation set and test set, which respectively contained 436 (1207 samples), 181 (541 samples), and 444 images (1228 samples). Parts of the data are shown in Figure 8.

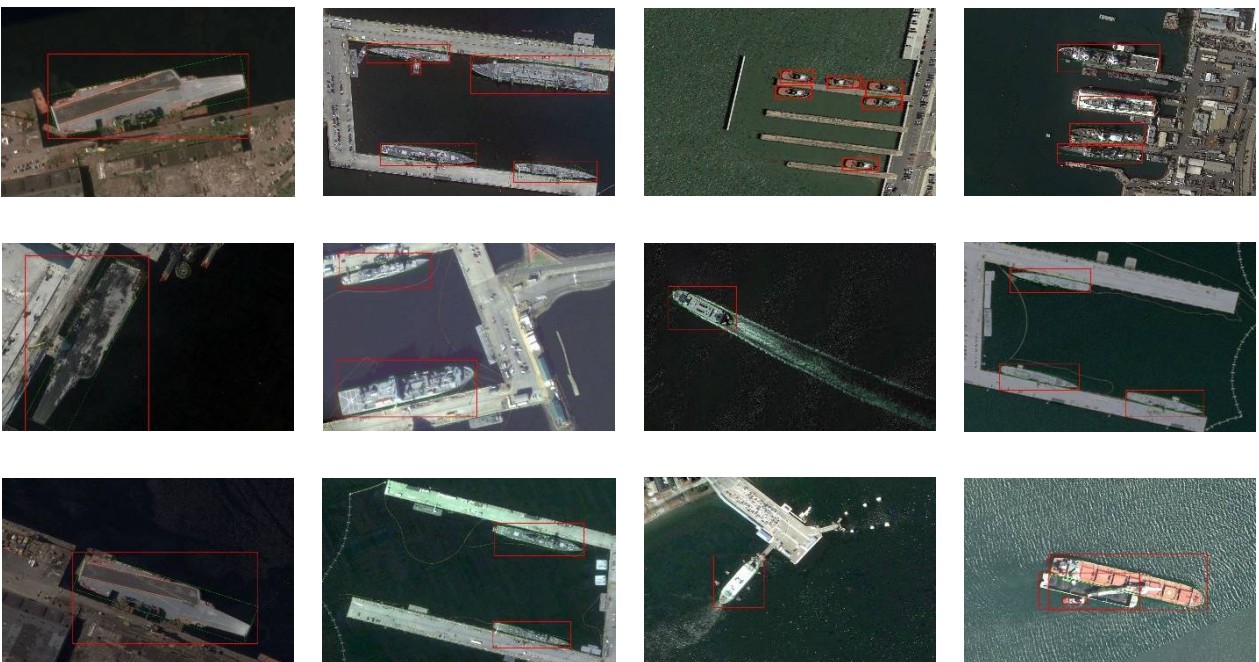

**Figure 8.** HRSC2016 samples. Training samples, validation samples, and test samples are displayed in the first, second and third rows, respectively.

### 3.2. Evaluation Metrics

Precision and recall rate are often used as evaluation criteria, which are defined as the ratio of the correct number of detected targets to the number of detected targets, and the ratio of the correct number of detected targets to the actual number of targets. The higher the precision and recall rate, the more accurate the detection performance of the model. However, in general, the precision rate is contradictory to the recall rate; that is, the recall rate is often low when the precision rate is high, and the accuracy rate is often low when the recall rate is high. Therefore, we added the F-measure index, which is a comprehensive indicator of imbalance between precision and recall rate. The calculation method is as follows:

$$Precision = \frac{TP}{TP + FP} \tag{5}$$

$$Recall = \frac{TP}{TP + FN} \tag{6}$$

$$F - measure = 2\frac{Precision * Recall}{Precision + Recall} \tag{7}$$

If the IoU between a prediction result and the ground truth value is higher than 0.5, it is defined as a true positive (*TP*). If IoU is lower than 0.5, it is defined as a false positive (*FP*). The actual target that is not detected is defined as a false negative (*FN*).

### 3.3. Implementation Details

For the experimental platform, we used an Intel Core i7-8700 3.20GHz six-core processor, with 32.0GB memory (Micron DDR4 2666MHz) and a graphics card with Nvidia GeForce RTX 2080 Ti 11GB video memory. In terms of software environment, we used the Windows10 Professional 64-bit operating system. The programming language used was Python, the deep learning framework used was Keras, and the TensorFlow framework as the backend was selected as the tool to build the model. The GPU computing platforms CUDA8.0 and cuDNN6.0 were used as the deep learning GPU acceleration library. We used the Adam optimization algorithm [33] to accelerate the training of our model. The size of the image data in the experiment was different, but it was uniformly resized to $512 \times 512$ before being input to the model.

### 3.4. Comparative Experiments of Different Methods

To test the effectiveness of SDGH-Net, we used YOLOv3 [17], Retinanet [34], EfficientDet [35] and faster R-CNN [13] networks as our experimental comparison methods. YOLOv3, Retinanet and EfficientDet are one-stage target detection algorithms, and faster R-CNN is a two-stage detection method.

YOLO series algorithms are known for their fast speed. Compared with the previous two versions of the algorithm, the YOLOv3 model has both drawbacks and improvements. A large part of the improvement of each generation of YOLO is determined by the improvement in the backbone network, from darknet-19 used in v2 to darknet-53 used in v3. Darknet-53 uses the residual network Residual and DarknetConv2D structure, the network is easier to optimize and thus the accuracy is higher.

Retinanet is a new target detection scheme proposed by Lin et al. at the same time as Focal Loss. Focal Loss is a loss scheme used to balance the positive and negative samples of the one-stage target detection method. The backbone network used by Retinanet is the Resnet network, which can effectively eliminate category imbalances and mine difficult samples with Focal Loss.

Based on EfficientNet [36], Google Brain proposed EfficientDet, a scalable model architecture for object detection. EfficientDet explores an effective FPN structure, proposes a bi-directional feature pyramid network (BiFPN), and performs Compound Scaling on each part of the detection network to improve model efficiency and accuracy.

Faster R-CNN is an excellent two-stage detection model that unifies candidate region generation and feature extraction, classification, and location refinement into a deep network framework. No calculations are repeated, which improves the running speed and detection accuracy.

The comparison between the proposed SDGH-Net and the other methods YOLOv3, Retinanet, EfficientDet, and faster R-CNN is quantitative and intuitive. Table 1 shows the precision, recall and F-measure of all methods. Figure 9 shows the visual prediction results of some test sets on different methods.

**Table 1.** Precision, recall, and F-measure of all methods.

| Methods | Precision (%) | Recall (%) | F-Measure (%) |
|---|---|---|---|
| YOLOv3 | 92.85 | 76.14 | 83.66 |
| Retinanet | 87.13 | 68.81 | 76.88 |
| EfficientDet | 91.25 | 69.63 | 78.98 |
| Faster R-CNN | 93.32 | 57.53 | 71.18 |
| SDGH-Net+TTA | 89.70 | 80.86 | 85.05 |

In terms of precision, the Faster R-CNN model achieved the highest score of 93.32%, followed by YOLOv3, and our SDGH-Net model achieved a good score of 89.70%. In terms of recall, the SDGH-Net model achieved the highest score of 80.86%, which is 23.33% higher than Faster R-CNN, which achieved the lowest score. YOLOv3, which achieved a score of 76.14%, ranked second. In terms of the comprehensive score F-measure, the SDGH-

Net model achieved the highest score of 85.05%. In short, YOLOv3 and SDGH-Net+TTA are better than the others but almost equally good, where YOLOv3 finds slightly more ships but also more false alarms. Figure 9 shows that SDGH-Net has the best detection performance.

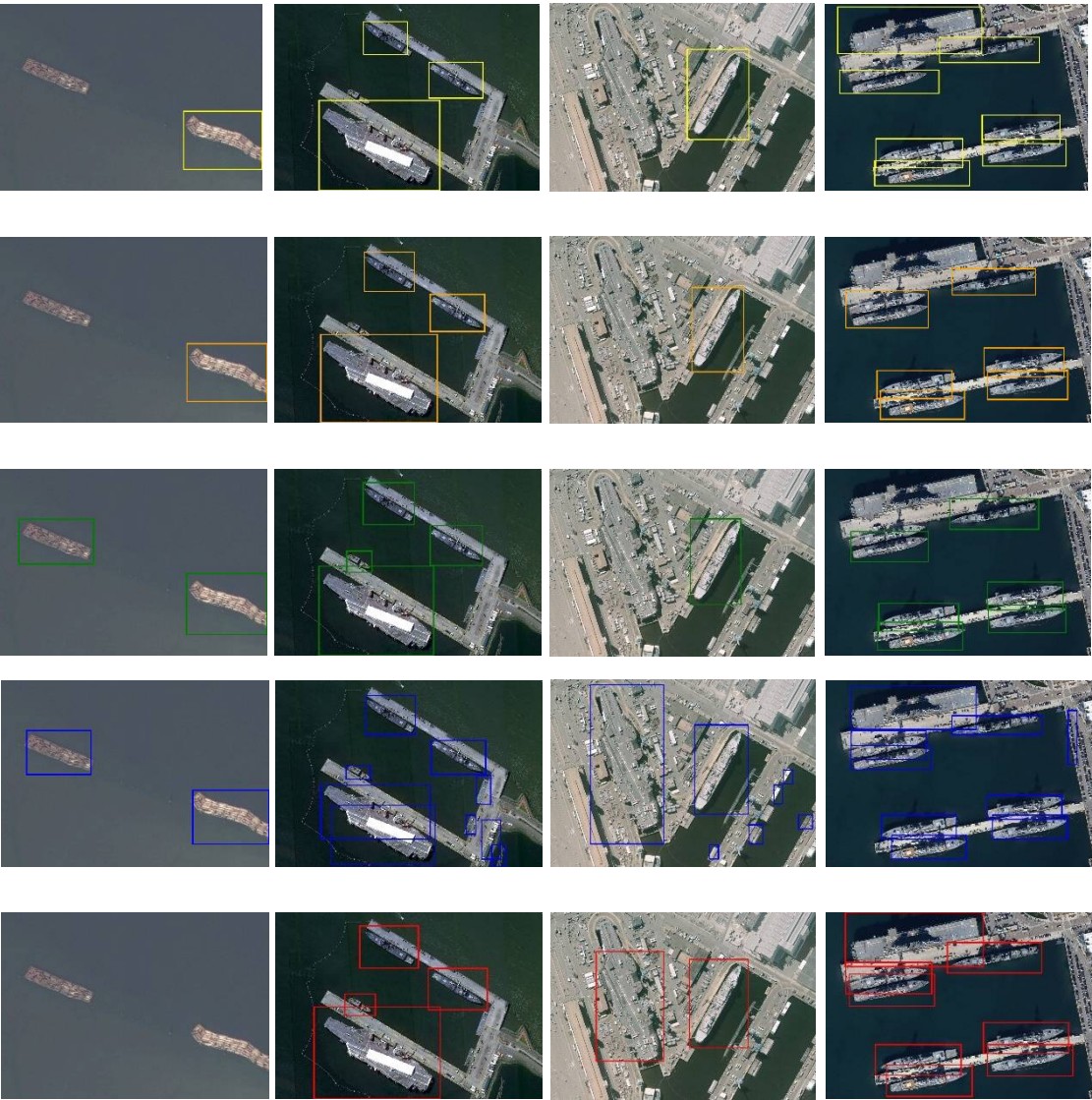

**Figure 9.** Comparison of test results of different methods. From top to bottom rows are YOLOv3, Retinanet, EfficientDet, Faster R-CNN and SDGH-Net.

With the need for real-time detection, model inference speed is becoming more important. We used the total inference time of 444 images (1228 samples) in the test set to evaluate the efficiency of the model. The total inference time includes model prediction time, post-processing time, and ship polygon annotation output time. The post-processing of SDGH-Net is the post-processing method described in the article, and the post-processing of other comparison models is the NMS post-processing. The number of parameters is usually reflected in the CPU time, so we increased the comparison of the number of parameters. The final efficiency analysis of each model is shown in Table 2. Although the reasoning efficiency of SDGH-Net is not as good as that of one-stage networks such as YOLOv3, Retinanet and EfficientDet, it is better than Faster R-CNN. Considering its good performance in precision, recall and F-measure, this inference efficiency is acceptable.

**Table 2.** Prediction time for test set of all methods.

| Methods | Backbone | Prediction Time (s) | Number of Parameters (MB) |
|---|---|---|---|
| YOLOv3 | Darknet-53 | 19.85 | 235 |
| Retinanet | ResNet-50 | 37.16 | 139 |
| Efficientdet | EfficientNet | 42.82 | 27 |
| Faster R-CNN | ResNet-50 | 174.02 | 108 |
| SDGH-Net+TTA | – | 139.58 | 115 |

### 3.5. Comparative Experiment of Different $T_{main}$ Values

$T_{main}$ is a key hyperparameter in our post-processing step. If the value of $T_{main}$ is too large, it is easy to miss the ships whose model are not certain. If the value of $T_{main}$ is too low, it is easy to cause only one polygonal annotation for a group of ships that are closer. To verify the correctness of the value of $T_{main}$ as 0.6 in our experiment, we conducted comparative experiments with the value of $T_{main}$ as 0.5 and 0.7. The evaluation index scores of the experiment are shown in Table 3, and part of the visualization results are shown in Figure 10. When the value of $T_{main}$ was 0.6, precision took second place, and both recall and F-measure achieved the highest scores. From the visualized results, the best result was obtained when $T_{main}$ was 0.6.

**Table 3.** Precision, recall, and F-measure of different $T_{main}$ values.

| The Value of $T_{main}$ | Precision | Recall | F-Measure |
|---|---|---|---|
| 0.5 | 90.67 | 79.97 | 84.98 |
| 0.6 | 89.70 | 80.86 | 85.05 |
| 0.7 | 86.34 | 78.25 | 82.10 |

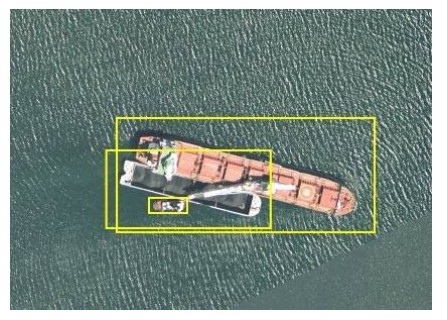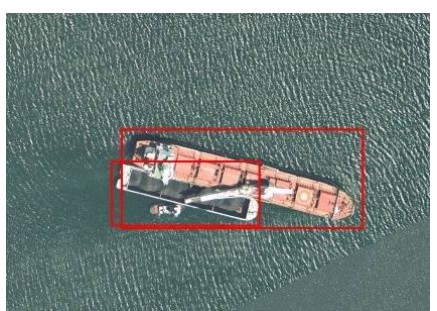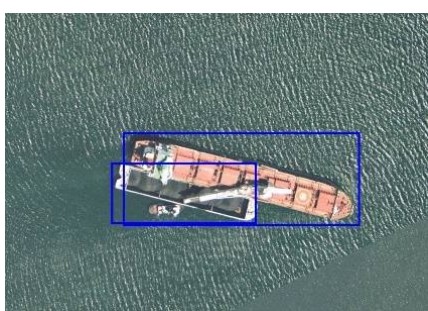

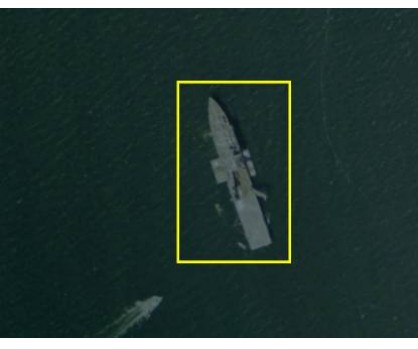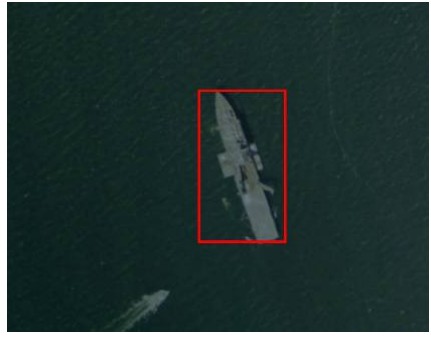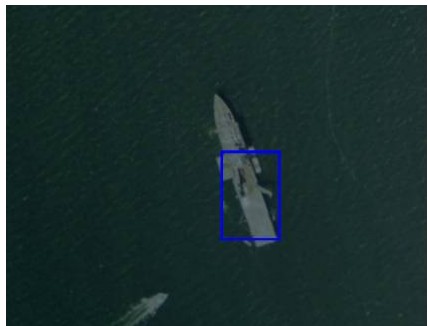

**Figure 10.** *Cont*.

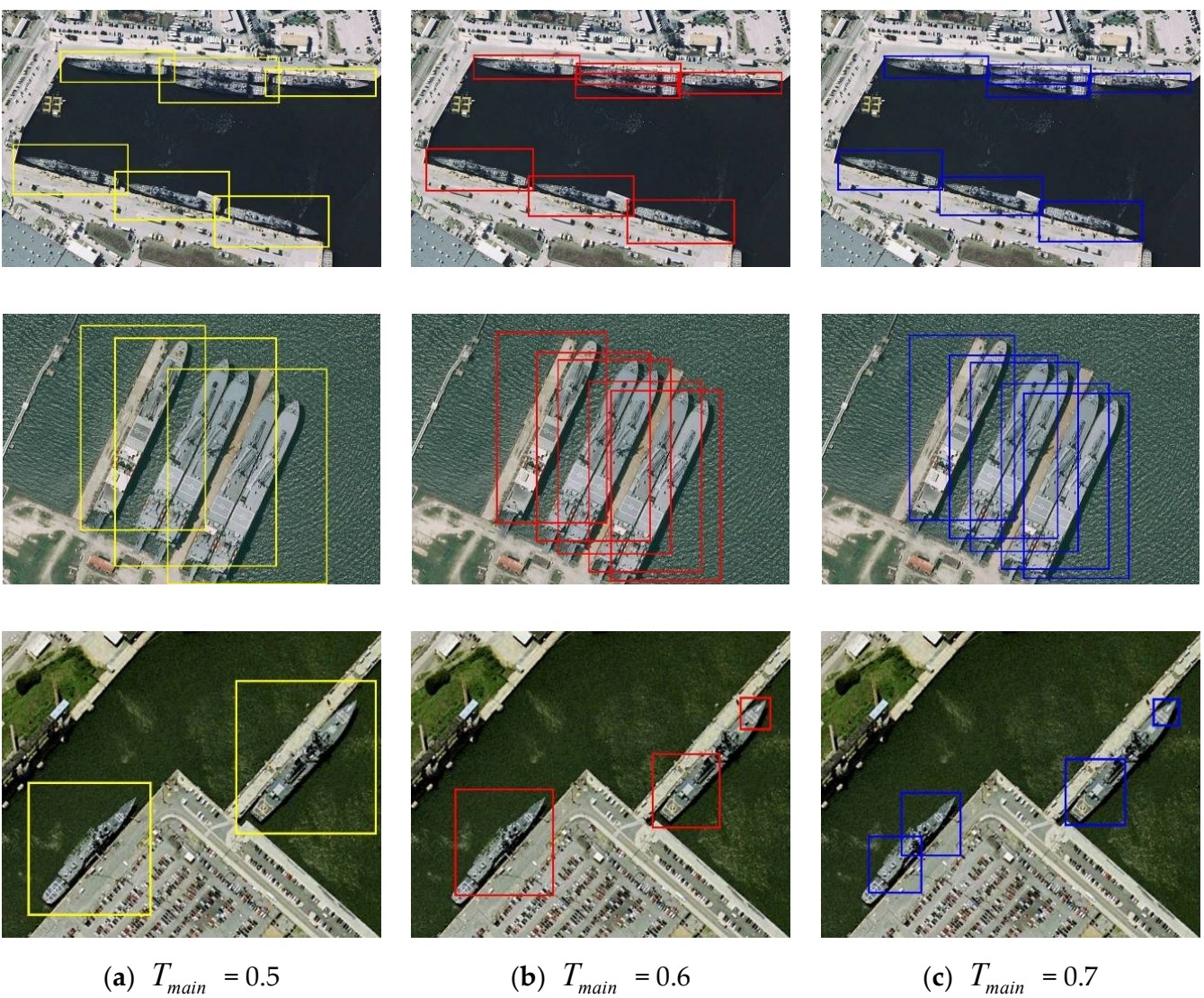

(**a**) $T_{main}$ = 0.5       (**b**) $T_{main}$ = 0.6       (**c**) $T_{main}$ = 0.7

**Figure 10.** Comparison of test results of different $T_{main}$ values. Left to right, columns correspond to $T_{main}$ = 0.5, $T_{main}$ = 0.6, and $T_{main}$ = 0.7. (**a**) $T_{main}$ = 0.5, (**b**) $T_{main}$ = 0.6, (**c**) $T_{main}$ = 0.7.

*3.6. Comparative Experiment with or without TTA*

We also performed test time augmentation (TTA) operations during the test. To verify the effectiveness of our TTA, we added a comparative experiment without TTA. The evaluation index score of the experiment and the test set prediction time are shown in Figure 11. The result showed that using TTA can improve the detection effect of the model, but it will also increase time consumption.

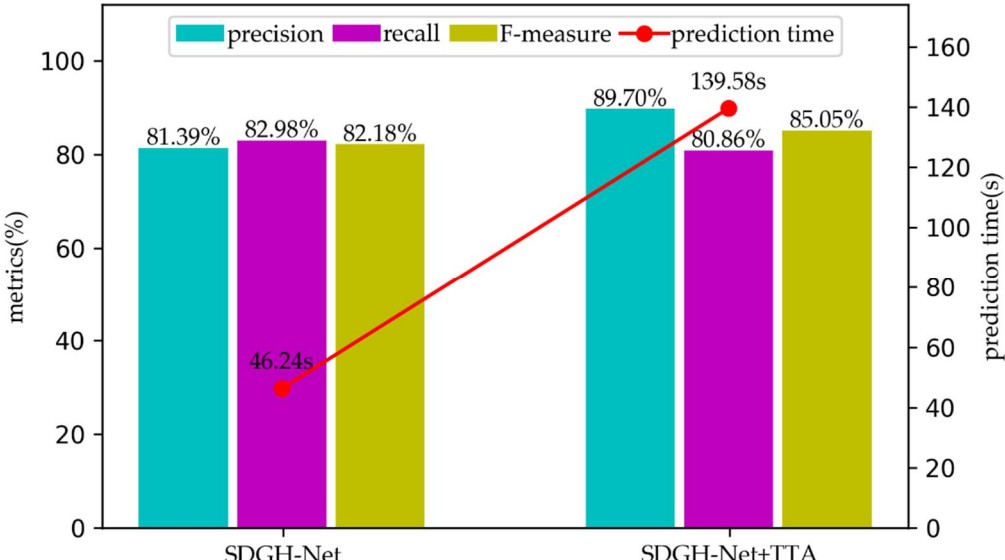

**Figure 11.** Metrics scores and prediction times of the comparative experiment with or without TTA. The blue column represents precision, the purple column represents recall, the yellow column represents F-measure, and the red dot represents the prediction time.

## 4. Discussion

Faster R-CNN and SDGH-Net both perform well for ships with similar backgrounds, but Faster R-CNN mistakenly detects many non-ship areas with similar ship characteristics, this is why Faster R-CNN has high precision but low recall rate. YOLOv3 and Retinanet, as well as EfficientDet, also have varying degrees of error and omission detection. In addition, SDGH-Net showed a more robust detection for side-by-side ships.

We also found that all models have a poor ability to detect ships in images taken in foggy weather, as shown in Figure 12. The foggy weather affects the characteristics of ships, which makes them difficult to detect. In subsequent experiments, images should be defogged first and then input into the network for ship detection.

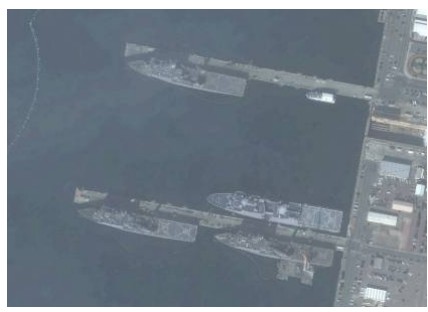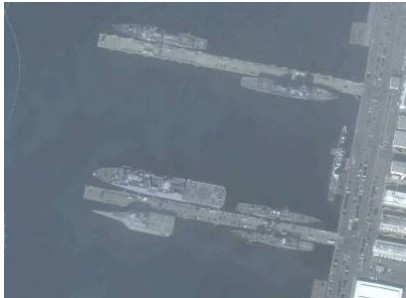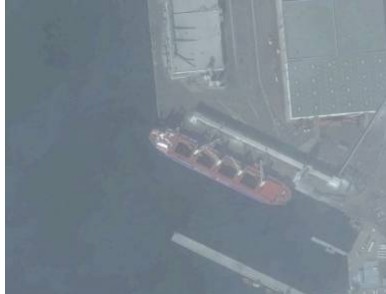

**Figure 12.** Images taken in foggy weather.

In the post-processing operation, the best result was obtained when the value of $T_{main}$ was 0.6 in this experiment. Figure 10 shows that too small a value easily leads to the detection of several ships that are close together as one ship, whereas too large a value leads to easily missing or separating ships with uncertain models. Therefore, the appropriate value of $T_{main}$ is crucial to the result.

From Figure 9, the precision of using TTA is 8.31% higher than that of not using TTA, indicating the obvious effect of TTA on precision. To explore the reasons for this, Figure 13 lists the ship feature map results of the comparative experiment and the ship polygon annotations obtained by post-processing. After using TTA, some inaccurate results that

did not use TTA were corrected. This is also why precision improved. The ship-like objects in Figure 13a are eliminated, the truncated ships in Figure 13b,c are connected, and the missed ships in Figure 13d are detected again.

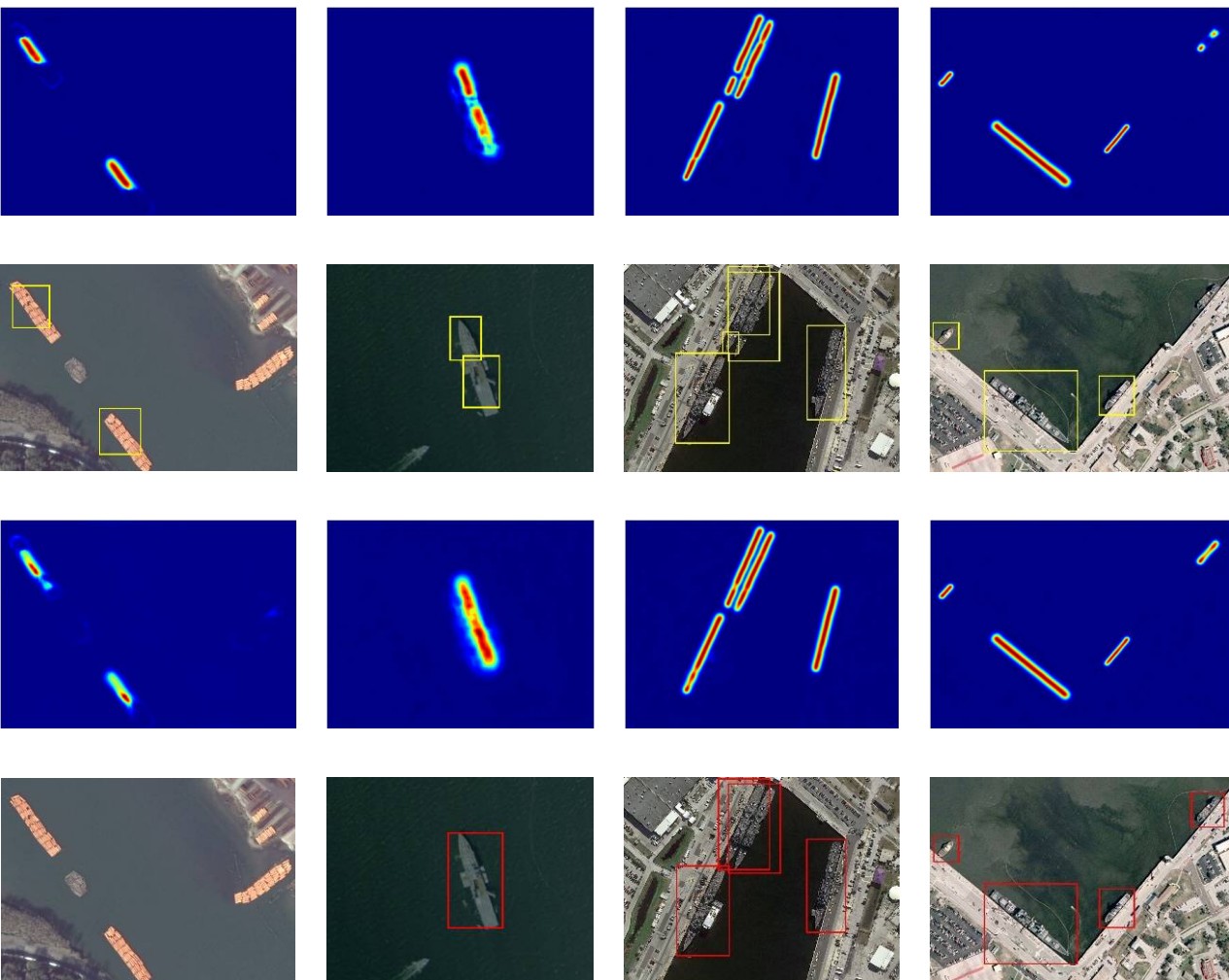

**Figure 13.** Examples of better results after using TTA. From top to bottom, the first and second rows represent the ship feature maps and polygon annotations obtained without using TTA, and the third and fourth rows represent the ship feature maps and polygon annotations obtained after using TTA.

Unfortunately, the recall using TTA is 2.12% lower than that without TTA. Examining the experimental results, we found that although TTA will correct some incorrect results, a few correct results will be processed incorrectly, as shown in Figure 14. However, this situation is rare, so the comprehensive evaluation index F-measure increased by 2.87%. When TTA was used, there were three more predictions than without TTA, so the prediction time was 93.43 s longer.

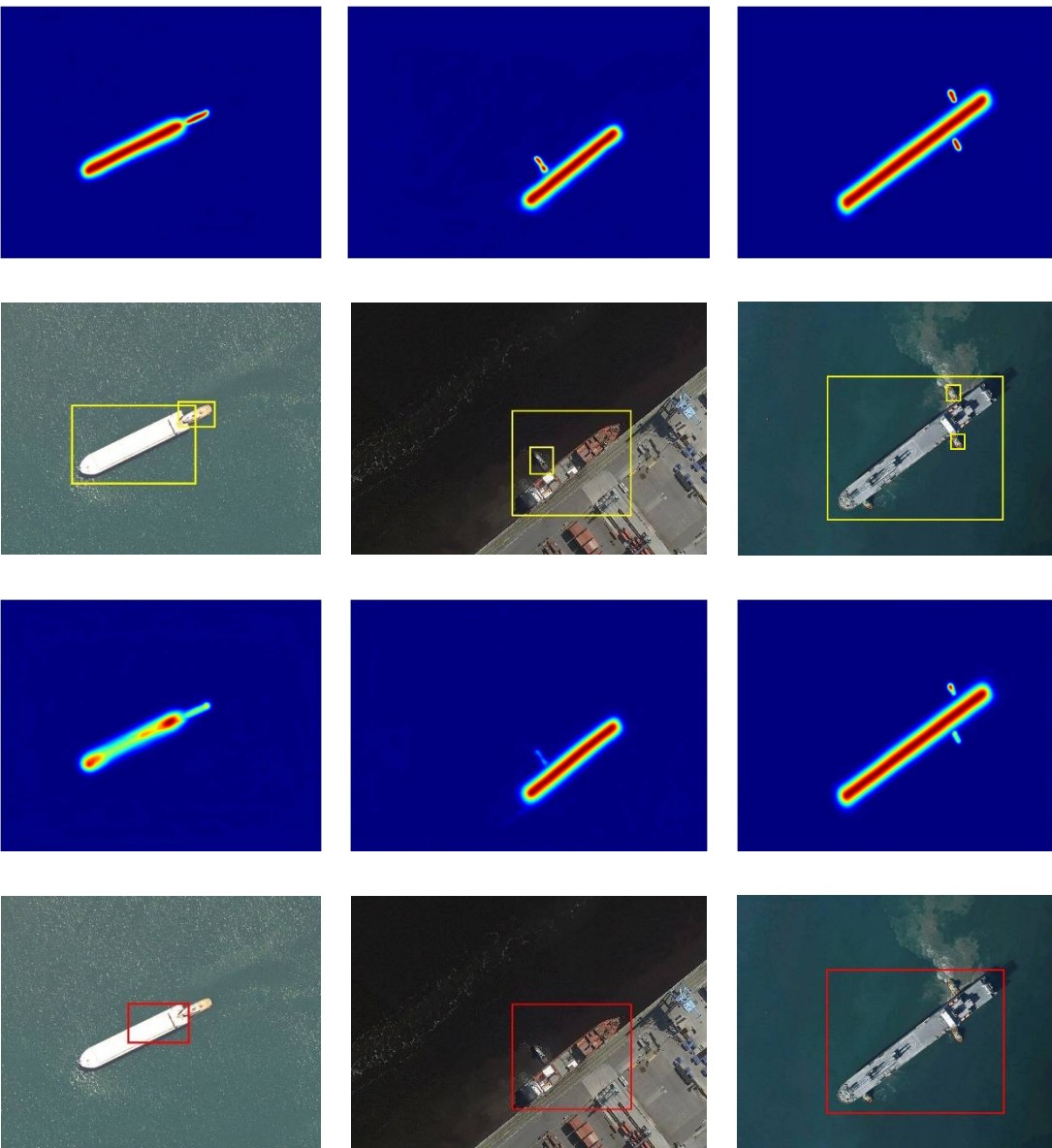

**Figure 14.** Examples of worse results after using TTA. From top to bottom, the first and second rows represent the ship feature maps and polygon annotations obtained without using TTA, and the third and fourth rows represent the ship feature maps and polygon annotations obtained after using TTA.

## 5. Conclusions

This paper proposes a novel and concise ship detection network, SDGH-Net, based on Gaussian heatmap regression. SDGH-Net is a fully convolutional model that avoids the loss of feature map spatial information due to the fully connected layer. SDGH-Net performs batch normalization processing after each layer of convolution to enhance the training speed of the model. To detect ships of different scales, hole convolutions of different rates were introduced and feature fusion in the form of addition is performed. To compensate for the imbalance of positive and negative samples in the data, we use the weighted mean square error as the loss function of the model. To further improve the generalization ability of the model, we adopted a test time augmentation strategy. Compared with other models, our method achieved high precision scores, and its recall and F-measure scores were the highest. In terms of efficiency, our method is superior to the two-stage target detection method Faster R-CNN.

In future work, our method has the following aspects worthy of further research and improvement: (1) The model structure could be improved. Although the model structure of SDGH-Net is relatively reasonable, there is still information loss. (2) A better loss function could be designed. The future loss function not only needs to balance positive and negative samples, but also needs to have the ability to mine difficult examples. (3) Post-treatment must be more appropriate. Our method obtains the ship polygon annotation from the ship characteristic heatmap, and more reasonable post-processing is crucial to the final result.

**Author Contributions:** Z.W. wrote the manuscript and designed the comparative experiments; Y.Z. and S.W. supervised the study and revised the manuscript; F.W. revised the manuscript and gave comments and suggestions to the manuscript; Z.X. assisted Z.W. in designing the architecture and conducting experiments. All authors have read and agreed to the published version of the manuscript.

**Funding:** This research was funded by the National Key R&D Program of China (Grant number: 2017YFB0504101, 2016YFC0803004).

**Institutional Review Board Statement:** Not applicable.

**Informed Consent Statement:** Not applicable.

**Data Availability Statement:** Not applicable.

**Acknowledgments:** The authors would like to thank the editors and the reviewers for their valuable suggestions.

**Conflicts of Interest:** The authors declare no conflict of interest.

## Abbreviations

The following abbreviations are used in this manuscript:

| | |
|---|---|
| SDGH-Net | Ship Detection Network based on Gaussian Heatmap Regression |
| SAR | Synthetic Aperture Radar |
| CNN | Convolutional Neural Networks |
| R-CNN | Regions with CNN Features |
| SPP-net | Spatial Pyramid Pooling Networks |
| SSD | Single Shot MultiBox Detector |
| SLC | Sequence Local Context |
| HRoI | Horizontal Region of Interest |
| BFP | Balanced Feature Pyramid |
| CF-SDN | Coarse-to-fine Ship Detection Network |
| PBS | Priority-based Selection |
| NMS | Non-Maximum Suppression |
| SPP | Spatial Pyramid Pooling |
| SSD | Single Shot Multibox Detector |
| YOLO | You Only Look Once |
| SCPs | Ship Center Points |
| BN | Batch Normalization |
| IoU | Intersection over Union |
| TP | True Positive |
| FP | False Positive |
| FN | False Negative |
| TTA | Test Time Augmentation |
| MSE | Mean Squared Error |
| Adam | Adaptive Moment Estimation |
| GPU | Graphics Processing Unit |

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
