# Peer review of "SDGH-Net: Ship Detection in Optical Remote Sensing Images Based on Gaussian Heatmap Regression"

_remotesensing, doi:10.3390/rs13030499_

Round 1

Reviewer 1 Report

The authors presents a ship detection methodology using optical remote sensing images based on the Gaussian heatmap regressions. The overall content of the document is well-written and suitable for this Journal. I do not have comments for the authors, only to once again review the paper before its final version.

Author Response

Response 1: Thank you very much for your recognition of this paper. We made some revisions to the paper. The revised part is marked in red.

Reviewer 2 Report

This work describes a slightly improved YOLO RNN based on gaussian heat maps. The analysis is worth while but needs further clarifications:

  1. In the abstract the claim that SDGH-Net "significantly improves..." is an exaggeration, when the numbers in table 1 are only slightly better than YOLOv3.
  2. A number of abbreviations are not explained in the introduction. Fx SDGH itself which is the basis of this work.
  3. It must be explained better, how the gaussian heat map precisely works and how it supposedly improves the results. Maybe in a simple pedagogical example?
  4. There a number of sentences which are hard to understand. Fx in line 90, why does the weight cause overfitting?
  5. Line 87: ability should probably be dis-ability.
  6. For comparing the various models,  the number of parameters is a good measure, which again often is reflected in cpu time.
  7. Line 264: how are nautical and offshore images defined?
  8. The authors might recap their results in Table 1 shows in short, that YOLOv3 and SDG-NET+TTA are better than the others but almost equally good, where YOOv3 finds slightly more ships but also more false alarms.
  9. It is not clear in lines 208-213, where the threshold Tmean precisely enters in the model?
  10. Fig 11: the prediction time in black on top of purple is unreadable.
  11. This work should refer to the work by Gang Tang et al. Remote Sensing 2020, 12, 4192. This is a parallel H-YOLO analysis of the same HRSC2016 dataset.

In summary, this work contains useful results for comparing YOLO type nets. Their SDGH-Net is only marginally better than YOLOv3, but lessons learned can be useful even when negative. The qualiity of the presentation and further explanations are, however, required.

Author Response

Point 1: In the abstract, the claim that SDGH-Net "significantly improves..." is an exaggeration, when the numbers in table 1 are only slightly better than YOLOv3.

Response 1: The expression "significantly improves" is inappropriate, we changed it to "improves".

Point 2: A number of abbreviations are not explained in the introduction. Fx SDGH itself which is the basis of this work.

Response 2: We explained some abbreviations in the introduction. They include convolutional neural networks (CNNs), regions with CNN features (R-CNN), spatial pyramid pooling networks (SPP-net), You Only Look Once (YOLO), single shot multibox detector (SSD), sequence local context (SLC), horizontal region of interest (HRoI), balanced feature pyramid (BFP), intersection over union (IoU)-balanced sampling (BS), rotational Libra R-CNN (R-Libra R-CNN), coarse-to-fine ship detection network (CF-SDN), priority-based selection (PBS), We designed SDGH-Net, which innovatively applies Gaussian heatmap regression method to ship detection in optical remote sensing images.

Point 3: It must be explained better, how the gaussian heat map precisely works and how it supposedly improves the results. Maybe in a simple pedagogical example?

Response 3: We added a description of how the Gaussian heat map works in line 98 of the paper: The overall process of Gaussian heatmap regression is relatively simple, and its framework is a fully convolutional neural network consistent with image segmentation. However, unlike image segmentation, the ground truth of the Gaussian heatmap regression is a Gaussian heatmap ranging from zero to one. Gaussian heatmap regression has no fully connected layer, and the output feature map has a large size and strong spatial generalization ability. In addition, using heatmap to supervise is an easier way to learn than regressing coordinates. The network does not need to convert the spatial position into coordinates by itself. Therefore, more tasks [26,27,28] use fully convolutional networks based on Gaussian heatmap regression for target detection.

Point 4: There a number of sentences which are hard to understand. Fx in line 90, why does the weight cause overfitting?

Response 4: We revised some sentences that were difficult to understand in line 96 of the paper: The weight of the fully connected layer depends heavily on the distribution of the training data, so the fully connected layer relatively easily causes overfitting.

Point 5: Line 87: ability should probably be dis-ability.

Response 5: We deleted the redundant sentence: The reason why the fully convolutional model has this ability is weight sharing.

Point 6: For comparing the various models,  the number of parameters is a good measure, which again often is reflected in cpu time.

Response 6: The number of parameters is usually reflected in the cpu time, so we increased the comparison of the number of parameters in line 350 of the paper.

Point 7: Line 264: how are nautical and offshore images defined?

Response 7: We have modified some descriptions of the dataset HRSC2016 in line 276 of the paper: The ships in the HRSC2016 [30] dataset include ships on sea and ships close inshore. There are a total of 1061 images including 70 sea images with 90 samples and 991 sea-land images with 2886 samples.

Point 8: The authors might recap their results in Table 1 shows in short, that YOLOv3 and SDG-NET+TTA are better than the others but almost equally good, where YOOv3 finds slightly more ships but also more false alarms.

Response 8: We added a summary of the results in Table 1 in line 342 of the article: In short, YOLOv3 and SDGH-Net+TTA are better than the others but almost equally good, where YOLOv3 finds slightly more ships but also more false alarms.

Point 9: It is not clear in lines 208-213, where the threshold Tmain precisely enters in the model?

Response 9: We re-described post-processing in line 218 of the paper: Because the SDGH-Net prediction result is a ship feature map, in order to obtain the ship polygon annotation, we need to use a post-processing method to extract the final polygon annotation from the ship feature map. If we directly take the area greater than 0 in the ship feature map as the ship area, it may cause the adhesion between close-range ship groups, and cause multiple ships to get only one polygon annotation. In order to overcome this problem, we carried out a simple treatment. As shown in Figure 5, first we perform threshold processing on the ship feature map, that is, the area larger than the threshold   is set as the main area of the ship, and then the main area is expanded to obtain the entire area of the ship. Finally, the boundary rectangle of the entire area of the ship is the boundary polygon we need. In this experiment, the value of Tmain was set to 0.6.

Point 10: Fig 11: the prediction time in black on top of purple is unreadable.

Response 10: We modified Figure 11 in line 378 of the article to make it more readable.

Point 11: This work should refer to the work by Gang Tang et al. Remote Sensing 2020, 12, 4192. This is a parallel H-YOLO analysis of the same HRSC2016 dataset.

Response 11: The work by Tang et al. is very useful to us and we referred it in line 87 of the article: Tang et al. proposed a single-shot ship detection approach based on region of interest preselected network. In order to efficiently match ship candidates, the principle of their approach is to distinguish suspected areas from the images based on hue, saturation, value (HSV) differences between ships and the background [24].

Reviewer 3 Report

Dear authors,
The work you have presented is hugely complex, and in its complexity, it must be well described.
I suggest to investigate the following aspects:

1) The algorithmic complexity needs a better explanation according to the specifications of the scene, such as, for example, the number of objects to be recognized present and the size of the image.

2) The implementation details are missing
2.1) data relating to the calculation times of the learning phase according to the images used
2.2) the calculation times after the learning process.

3) The database you are referring to is not accessible. Kindly provide a copy of it if possible or a detailed description of its content.

5) How does ship size normalization take place?

6) Concerning paragraph 2.2.2, please explain in more detail how the algorithm chose the optimal matrix size.

7) Paragraph 3.5 talks about checking the correctness of Tmain. Kindly prove that this is a correct value.

8) The paper must present all acronyms before using them, such as BN. Please proofread the text.

Best Regards

Author Response

Point 1: The algorithmic complexity needs a better explanation according to the specifications of the scene, such as, for example, the number of objects to be recognized present and the size of the image.

Response 1: We added a description of the size of the image of the input model in line 306, and a description of the number of objects to be recognized in line 345:

The size of the image data in the experiment was different, but it was uniformly resized to 512 × 512 before being input to the model.

We used the total inference time of 444 images (1228 samples) in the test set to evaluate the efficiency of the model.

Point 2: The implementation details are missing

2.1) data relating to the calculation times of the learning phase according to the images used

2.2) the calculation times after the learning process.

Response 2: We added the implementation details for calculating the total inference time in line 347:

The total inference time includes model prediction time, post-processing time, and ship polygon annotation output time. The post-processing of SDGH-Net is the post-processing method described in the article, and the post-processing of other comparison models is the NMS post-processing.

Point 3: The database you are referring to is not accessible. Kindly provide a copy of it if possible or a detailed description of its content.

Response 3: The database we referred is HRSC2016, and the corresponding referred papers explain the data in detail. The download address http://www.escience.cn/people/liuzikun/DataSet.html provided in the paper is no longer valid. We saved the backup of Baidu cloud disk, the link is https://pan.baidu.com/s/1BO2UX6dICVa33qUqIMCFrw , and the extraction code is shbp.

Point 4:How does ship size normalization take place?

Response 4: Ship size normalization is not done in our paper.

Point 5:Concerning paragraph 2.2.2, please explain in more detail how the algorithm chose the optimal matrix size.

Response 5: Concerning paragraph 2.2.2, the rate of the dilated convolution is a hyperparameter. The value of 2, 4, and 8 is because these three numbers are mostly used in the industry. This is like the size of the convolution kernel, we generally take the value 3×3. In summary, Rate values 2, 4, and 8 are empirical values. And the experiment proves their effectiveness.

Point 6:Paragraph 3.5 talks about checking the correctness of Tmain. Kindly prove that this is a correct value.

Response 6: We re-described how Tmain works in paragraph 2.3 and added a discussion of   in paragraph 3.5:

Because the SDGH-Net prediction result is a ship feature map, to obtain the ship polygon annotation, we need to use a post-processing method to extract the final polygon annotation from the ship feature map. If we directly take an area greater than 0 in the ship feature map as the ship area, it may cause adhesion between close-range ship groups, and cause multiple ships to be assigned only one polygon annotation. To overcome this problem, we applied a simple treatment. As shown in Figure 5, we performed threshold processing on the ship feature map; that is, the area larger than the threshold Tmain was set as the main area of the ship, and then the main area was expanded to obtain the entire area of the ship. Then, the boundary rectangle of the entire area of the ship was the boundary polygon we needed. In this experiment, the value of Tmain was set to 0.6.

If the value of Tmain is too large, it is easy to miss the ships whose model are not certain. If the value of Tmain is too low, it is easy to cause only one polygonal annotation for a group of ships that are closer.

Point 7:The paper must present all acronyms before using them, such as BN. Please proofread the text.

Response 7: We checked all acronyms and made corrections. They include convolutional neural networks (CNNs), regions with CNN features (R-CNN), spatial pyramid pooling networks (SPP-net), You Only Look Once (YOLO), single shot multibox detector (SSD), sequence local context (SLC), horizontal region of interest (HRoI), balanced feature pyramid (BFP), intersection over union (IoU)-balanced sampling (BS), rotational Libra R-CNN (R-Libra R-CNN), coarse-to-fine ship detection network (CF-SDN), priority-based selection (PBS), We designed SDGH-Net that innovatively applied the method of Gaussian heat map regression to ship detection in optical remote sensing images.

Round 2

Reviewer 2 Report

The authors have responded satisfactorily to most comments, and I recommend publication.

Author Response

Point 1: The authors have responded satisfactorily to most comments, and I recommend publication.

Response 1: Thank you very much for your valuable comments and approval of the revised article.

Reviewer 3 Report

Dear Authors,

I am sorry to write to you that your answers to my questions are not exhaustive. For example, you did not answer the following questions exhaustively:

1) The algorithmic complexity needs a better explanation according to the specifications of the scene, such as, for example, the number of objects to be recognized present and the size of the image.

2) The implementation details are missing
2.1) data relating to the calculation times of the learning phase according to the images used
2.2) the calculation times after the learning process.

3) The database you are referring to is not accessible. Kindly provide a copy of it if possible or a detailed description of its content.

The paper is rejected.

Best Regards

Author Response

Point 1: The algorithmic complexity needs a better explanation according to the specifications of the scene, such as, for example, the number of objects to be recognized and the size of the image.

Response 1: Thank you for your suggestions. We have described the number of objects to be recognized present. As mentioned in line 345 of the paper, our test set has a total of 444 images, which contain 1228 objects.

We have described the size of the image in line 306 of the paper. These 444 images’ size was uniformly resized to 512 × 512 before being input to the network. Because of the resize process, the original image size is meaningless, and we not described the original image size.

Original text on line 345:

We used the total inference time of 444 images (1228 samples) in the test set to evaluate the efficiency of the model.

Original text on line 306:

The size of the image data in the experiment was different, but it was uniformly resized to 512 × 512 before being input to the model.

Point 2: The implementation details are missing

2.1) data relating to the calculation times of the learning phase according to the images used

2.2) the calculation times after the learning process.

Response 2: We gratefully appreciate for your suggestion. We added the implementation details for calculating the total inference time in line 347.

In the field of computer vision (CV), the index to evaluate the efficiency of the target detection model is frames per second (FPS), that is, how many pictures the model can detect per second. Evaluating the efficiency of a model generally only looks at the efficiency of the prediction stage, and does not pay much attention to the training time. Because when we deploy the model to the target detection workstation, the model is already trained. The evaluation index we use is the total inference time includes model prediction time, post-processing time, and ship polygon annotation output time. The post-processing of SDGH-Net is the post-processing method described in the article, and the post-processing of other comparison models is the NMS post-processing.

Original text on line 347:

The total inference time includes model prediction time, post-processing time, and ship polygon annotation output time. The post-processing of SDGH-Net is the post-processing method described in the article, and the post-processing of other comparison models is the NMS post-processing.

Point 3: The database you are referring to is not accessible. Kindly provide a copy of it if possible or a detailed description of its content.

Response 3: The database we referred is HRSC2016, and the corresponding referred papers [1] explain the data in detail:

The HRSC2016 dataset contains images from two scenarios including ships on sea and ships close inshore. All the images are collected from six famous harbors. We not only collect the default images shown by Google Earth, but also download the history images in the same place. The image resolutions are between 2-m and 0.4-m. The image sizes range from 300×300 to 1500×900 and most of them are larger than 1000×600.

In the process of collecting data, we recorded the image information including harbor, data source, image date, geographic coordinate, resolution layer, scale, etc. It is worth noting that Google Earth’s geographic coordinate system may assign slightly different coordinates to the same location on the earth. And we just recorded a near geographic coordinate for each image.

We get 1061 images including 70 sea images with 90 samples and 991 sea-land images with 2886 samples. After adding annotations to these samples, we split the dataset into training, validation and test set which contains 436 images including 1207 samples, 181 images including 541 samples and 444 images including 1228 samples respectively.

Most of the HRSC2016 dataset images are inshore data. In order to satisfy the needs of the works for ship detection on sea, we provide another 610 images from Mumansk harbor, including 156 sea images and 454 sea-land images but without annotations. In the future, we will further extend our dataset.

Unfortunately, the download address (http://www.escience.cn/people/liuzikun/DataSet.html) provided in this paper is no longer valid. We saved the backup of Baidu cloud disk, the link is https://pan.baidu.com/s/1BO2UX6dICVa33qUqIMCFrw , and the extraction code is shbp. If you want to view the data set, you can view it through Baidu Cloud Disk.

Although the download link provided in the paper is invalid, we still need to refer this paper according to the reference rules.

  1. Liu, Z.; Yuan, L.; Weng, L.; Yang, Y. A high resolution optical satellite image dataset for ship recognition and some new baselines. In Proceedings of the 6th International Conference on Pattern Recognition Application and Methods (ICPRAM 2017). Porto, Portugal, 24–26 February 2017; pp. 324–331.
